# Numerical study of the error sources in the experimental estimation of thermal diffusivity: an application to debris-covered glaciers

Calvin Beck[1,2] and Lindsey Nicholson[2]

[1]Normandie Université – UNICAEN - UNIROUEN, CNRS, UMR 6143 M2C, Laboratoire Morphodynamique Continentale et Côtière, Caen, France
[2]Department of Atmospheric and Cryospheric Sciences, University of Innsbruck, Innsbruck, Austria
**Correspondence:** Calvin Beck (calvin.beck@unicaen.fr)

**Abstract.** A surface debris layer significantly modifies underlying ice melt dependent on the thermal resistance of the debris cover, with thermal resistance being a function of debris thickness and effective thermal conductivity. Thus these terms are required in models of sub-debris ice melt. The most commonly used method to calculate effective thermal conductivity of supraglacial debris layers applies heat diffusion principles to a vertical array of temperature measurements through the supraglacial debris cover combined with an estimate of volumetric heat capacity of the debris as presented by Conway and Rasmussen (2000). Application of this approach is only appropriate if the temperature data indicates that the system is predominantly conductive and even in the case of a pure conductive system, the method necessarily introduces numerical errors that can impact the derived values. The sampling strategies used in published applications of this method vary in sensor precision, and spatio-temporal temperature sampling strategies, hampering inter-site comparisons of the derived values, and their usage at unmeasured sites. To address this, we use synthetic datasets to isolate the numerical errors of the temporal and spatial sampling interval, and the precision of sensor temperature and position in recovering known thermal diffusivity values using this method. On the basis of this we can establish a sampling an analytical strategy to minimize the methodological errors. Our results show that increasing temporal and spatial sampling intervals increase truncation errors and systematically underestimate calculated values of thermal diffusivity. The thermistor precision, the shape of the diurnal temperature cycle, the debris thermal diffusivity and misrepresenting the vertical thermistor position also result in systematic errors, that show strong cross-dependencies dependent on signal to noise ratio with which spatio-temporal temperature gradients are captured. We provide an interactive analysis tool and best-practice guidelines to help researchers investigate the effect of the sampling interval on calculated sub-debris ice melt and plan future measurement campaigns. These findings can be used to plan optimal field sampling strategies for future campaigns and as a guide for common reanalysis of existing datasets to allow intercomparison across sites.

## 1 Introduction

Debris-covered glaciers can be found in tectonically active mountain regions such as Alaska, the European Alps, High Mountain Asia, or New Zealand (Herreid and Pellicciotti, 2020), where large amounts of debris migrate into the ice via glacial and periglacial processes (Shugar and Clague, 2011; Scherler et al., 2018; Anderson et al., 2018). Debris falling onto the ablation zone contributes directly to any surface debris load, while debris added to the glacier surface in the accumulation zone or

sourced subglacially is transported englacially to the ablation area of the glacier, where it melts out and contributes additional debris load (Nicholson and Benn, 2006; Kirkbride and Deline, 2013; Anderson et al., 2018), as shown in Figure 1a. In comparison to clean ice, thin or patchy debris amplifies ice melt due to its higher absorptivity of short-wave radiation while thicker debris layers reduce ice melt due to the insulation and attenuation of the diurnal heating signal (Inoue and Yoshida, 1980; Kayastha et al., 2000; Kirkbride and Dugmore, 2003; Mihalcea et al., 2006; Brock et al., 2010; Reznichenko et al., 2010; Fyffe et al., 2014; Minora et al., 2015). The relationship between debris thickness and ablation rate varies for different debris layer compositions and prevailing climatological conditions but retains the same character (Fig. 1b). The critical debris thickness beyond which sub-debris ice ablation is inhibited compared to clean ice ablation ranges from 15 to 115 mm (Østrem, 1959; Mattson, 1993; Nicholson and Benn, 2006) dependent on the debris optical and thermal properties and the ambient climate (Inoue and Yoshida, 1980; Nakawo and Takahashi, 1982; Adhikary et al., 1997; Reznichenko et al., 2010). Therefore, in contrast to clean ice glaciers, where the melt increases towards the glacier tongue in response to typical environmental temperature lapse rates, the spatial pattern of melt of debris-covered glaciers depends more on the debris thickness than on the elevation (e.g. Benn et al., 2012; Rowan et al., 2021; Nicholson et al., 2021). Herreid and Pellicciotti (2020) found that $7.3 \pm 3.3\%$ of all mountain-glacier area is covered by a rock debris cover, which at a global scale delays the loss of debris-covered glaciers for the coming decades (Rounce et al., 2023). With continued glacier decline debris-covered glacier surfaces are expected to increase in absolute and percentage terms in the future (Deline and Orombelli, 2005; Kellerer-Pirklbauer et al., 2008; Quincey and Glasser, 2009; Bhambri et al., 2011; Bolch et al., 2012; Kirkbride and Deline, 2013; Thakuri et al., 2014; Scherler et al., 2018; Tielidze et al., 2020), highlighting the need for accurate modeling of sub-debris ice melt to be included in future glacier projections (Rounce et al., 2015a).

Although under certain circumstances heat can be transferred through the debris by convection, advection, and radiation, observations (e.g. Conway and Rasmussen, 2000; Nicholson and Benn, 2012) show that the system often, and especially under dry stable meteorological conditions, approximates Fourier's law of conduction $q = -k\nabla T$, where $q$ represents the local heat flux density, $k$ the thermal conductivity, and T the temperature (Fourier, 1955; Cannon, 1984). Consequently, in models of glacier ice melt, the energy supply for ice melt beneath the debris cover is typically treated as if it were heat conduction only (e.g. Reid and Brock, 2010; Fyffe et al., 2014), driven by the surface temperature, debris thickness and a value of debris thermal conductivity to be supplied as a model parameter. As a second consequence, Fourier's law of conduction has also been used to derive representative parameter values of effective debris thermal conductivity for horizontally homogeneous debris layers from field observations of spatio-temporal variations in debris temperature. To do this, the one-dimensional heat conduction equation for a homogeneous, isotropic medium (Eq. 1) is used to derive the apparent thermal diffusivity, $\kappa$, from the spatio-temporal variation of a vertical profile of temperature measurements, by finding the gradient of the regression line between the first derivative of temperature with time and the second derivative of temperature with depth (Conway and Rasmussen, 2000). Effective thermal conductivity $k$ can then be calculated from $\kappa$ and the volumetric heat capacity of the debris, given by the specific heat capacity $c_s$ and the material density $\rho$ (Eq. 2), including the porosity for a granular material.

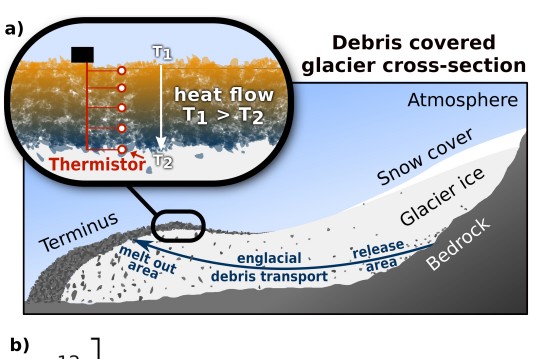

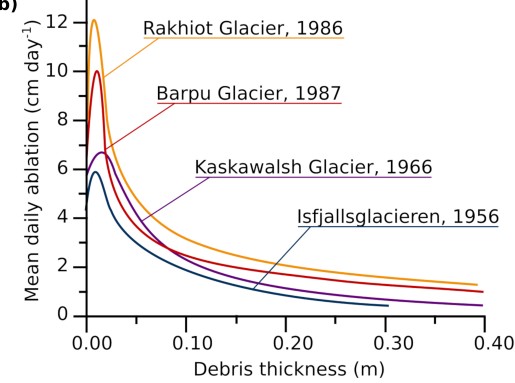

**Figure 1.** a) Schematic of a debris-covered glacier with debris transport of subglacially sourced rock debris from release area to melt out area. The inset shows a classical thermal diffusivity measurement site, consisting of thermistors at several heights between the near-surface and the debris-ice interface. b) Measurements of the so-called Østrem curves for different glaciers show a common pattern of variation of daily melt rate versus the debris depth, with site-specific variations in maximum ablation and the debris thickness associated with it. Redrawn from Mattson (1993).

$$\frac{\partial T}{\partial t} = \kappa \frac{\partial^2 T}{\partial x^2} + const. \tag{1}$$

$$\kappa = \frac{k}{\rho \cdot c_s} \quad \begin{array}{l} \leftarrow \text{thermal conductivity} \\ \leftarrow \text{heat capacity} \end{array} \tag{2}$$

60

Application of this method therefore requires (1) a vertical array of temperature measurements through the supraglacial debris cover (Fig. 1a) for conditions in which the debris heat transfer closely approximates that of a conductive system from which the apparent $\kappa$ is derived and (2) an estimate of the volumetric heat capacity of the debris used to convert the apparent $\kappa$ into effective conductivity.

65

To meet the first requirement a sample site must be chosen for which lateral heat transfer can reasonably be expected to be negligible; so a site that is horizontally homogeneous in factors such as slope, debris type, thickness and without evidence of any hydrological heat transfer. Then the observed temperatures must be evaluated to find specific time periods and vertical subsets of debris temperature profile data that are identified as being 'well-behaved' approximations of a conductive system; data that show evidence of non-conductive processes can be excluded from subsequent analysis (Conway and Rasmussen, 2000). To meet the second requirement, estimates of the debris porosity, rock density thermal properties must be made. Commonly used values for these terms are porosity of $0.3$, rock density of $2700 \, \mathrm{kg \, m^{-2}}$ and rock specific heat capacity of $750 \, \mathrm{J \, kg^{-1} K^{-1}}$, with a $10\%$ error applied to the combined terms (Conway and Rasmussen, 2000). Most studies assume that the pore spaces are air filled when calculating the volumetric heat capacity, but in principle, if the debris cover is known to be fully saturated, a water-filled case can be used to obtain the volumetric heat capacity of the sampled debris layer (Nicholson and Benn, 2006). In an ideal case, this workflow can yield a reliable estimate of effective thermal conductivity from a homogenous dry portion of the debris with stable meteorological forcing conditions and minimal non-conductive processes. Onward use of these effective dry debris thermal conductivity data in surface energy balance models can allow for non-conductive processes and non-uniform debris layers to be included in the model structure by, for example, accounting for stratification in the debris porosity and air flow through the debris (Evatt et al., 2015), stratification of moisture content and associated phase changes within the debris layer (Collier et al., 2014; Evatt et al., 2015; Giese et al., 2020).

As natural debris covers often show vertical variation in porosity, grain size and moisture content, recent studies have explored multi-layered applications of the thermal diffusion representation of the debris layer. Laha et al. (2022) perform multiple rather than single regression analysis to account for (i) unknown depth variation in $\kappa$ in a two-layer model and (ii) non-conductive heat sources/sinks. They apply various methods to synthetic datasets to highlight that applying the original method of Conway and Rasmussen (2000) produces large errors when trying to recover a target $\kappa$ that varies with depth and that unequally spaced temperature measurements introduce substantial truncation errors. If unequal spacing of measurements cannot be avoided, their new Bayesian method of determining $\kappa$ outperforms that of Conway and Rasmussen (2000). Petersen et al. (2022) also included a term for depth varying $\kappa$ into the heat conduction equation and perform multiple linear regression to solve for its variation with depth in natural debris cover, identifying non-conductive processes as the residual from a comparison of the observed and modeled time-dependent temperature evolution. They find non-negligible heat transfer related air motion and latent heat fluxes within the debris on Kennicott Glacier. These approaches offer solutions for the potential of vertically varying debris properties and allow quantified assessment of non-conductive processes in measured field sites.

Despite these new developments, the method of Conway and Rasmussen (2000) has been historically widely used (e.g. Nicholson and Benn, 2006; Haidong et al., 2006; Juen et al., 2013; Chand and Kayastha, 2018; Rounce et al., 2015a; Rowan et al., 2021), and has provided the majority of published debris thermal conductivity values used in generalized surface energy balance models (Reid and Brock, 2010; Fyffe et al., 2014; Evatt et al., 2015), and for regional intercomparisons of supraglacial debris properties (Fontrodona-Bach et al., 2025). Thus, many studies of debris covered glaciers rely upon the robustness of debris thermal properties produced following Conway and Rasmussen (2000). The limited number of datasets used to provide generalized values of effective thermal conductivity have deployed very different field and analytical strategies, with

temporal and spatial sampling intervals, thermistor placement within the debris, debris depth of the sampled site, and sensor precision all selected *ad hoc* in different studies and differing from measurement site to measurement site (e.g. Juen et al., 2013; Chand and Kayastha, 2018; Rowan et al., 2021). For example, spatio-temporal temporal sampling intervals range from two to tens of centimetres and from five minutes to six hours, sometimes including time-averaged rather than sampled temperatures

(Appendix B). The impact of these choices on the derived $\kappa$ values are not well addressed in the published literature, but for example, the same data from Imja glacier in Nepal analyzed at 30 minute (Rounce et al., 2015a) and 60 minute (Rowan et al., 2021) intervals yielded thermal conductivity values that differed by almost $40\%$ despite using the same properties to derive thermal conductivity from $\kappa$. This highlights that baseline literature values that are used in surface energy balance modeling may be differently influenced by sensor, installation and numerical truncation errors, and indicates that care should

be taken when comparing across sites for which different instrumental and analytical choices have been made (e.g. Rowan et al., 2021; Miles et al., 2022). Therefore, a deeper exploration of the error sources of this method is warranted, and it would be advantageous to develop a standardized field and analytical implementation strategies.

## 2   Aim of this study

This study explores the effect of measurement setup on $\kappa$ values derived using the method of Conway and Rasmussen (2000)

in order to highlight the potential dependency of published values of thermal conductivity on the spatiotemporal intervals chosen for the analysis, and on the sensor precision and locational accuracy. To achieve this we apply the method of Conway and Rasmussen (2000) to data generated using a forward diffusivity model for a purely conductive system with a specified value of $\kappa$, and assess how closely the known $\kappa$ is recovered when varying choices of instrumental and analytical setups. Since the approach recommended by Conway and Rasmussen (2000) is only valid for conductive systems, we focus our

study on a purely conductive system to provide a baseline reference for individual method-related error sources, expanding the analysis of the impact of irregular spacings performed in Laha et al. (2022) to include assessment of a wider range of field measurement choices. By isolating the individual roles of these different error sources, they can be quantified, and their tendencies can be understood, thereby making a more critical reassessment of the extent to which differences in published effective thermal conductivity values reflect real world differences in debris properties or instrumental and analytical choices

possible. We provide an interactive tool (https://github.com/calvinbeck/TC-DTD) to allow analysis of the combined errors for any given measurement procedure and a best practice guideline on how to minimize the systematic errors of using this method (Appendix A).

## 3   Methods

### 3.1   Artificial data for benchmarking derived thermal diffusivity

To test the method of Conway and Rasmussen (2000) for different scenarios, we generate synthetic data for debris cover thicknesses of 30 cm and 100 cm and $\kappa$ values of $5 \cdot 10^{-7}\,m^2\,s^{-1}$, and $10 \cdot 10^{-7}\,m^2\,s^{-1}$, to represent a range of values obtained from

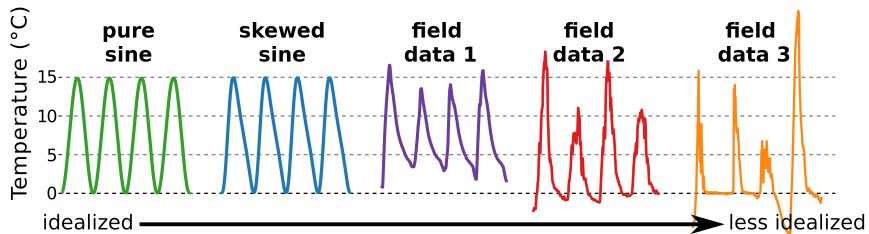

**Figure 2.** Characteristics of the surface temperature forcing for the artificial data generation, which consists of 10-day timeseries of two analytical sine curves and three experimental temperature measurements within the debris layer. The sine curves have a average temperature of $7.5\,°$C and the same amplitude. Surface forcing from field data is derived from the upper most thermistor which lies $1-5\,$cm below the surface, as indicated in brackets. Field data 1 and 3 was recorded at Lirung glacier (Nepal) during September 2013 ($5\,$cm below surface) and April 2014 ($1\,$cm below surface) respectively and was by provided by Chand and Kayastha (2018). Field data 2 was recorded at Vernagtferner (Austria) during June 2010 ($4\,$cm below surface) and was provided by Juen et al. (2013). The color scheme of these forcings is used in subsequent figures.

previous field studies from glaciers across the globe (Laha et al., 2022). The interactive tool allows users to perform analyses for any alternative choice of debris thickness and $\kappa$. To generate data for a perfectly conductive system, we force the heat equation with five 10-day surface temperature time-series (Fig. 2) and a $0\,°$C boundary condition for the debris ice interface. The first two days of temperature forcing data are used to initialize the model, and the different debris layer thicknesses are represented by varying the number of vertical grid points in the domain while maintaining equidistant spacing.

We use the Crank and Nicolson (1947) method to solve the heat conduction equation for this set of given constraints. This implicit finite difference method is convergent second-order in time and numerically stable. The method is based on the trapezoidal rule and is a combination of the Euler forward and backward methods in time. For the thermal heat equation, it results in the following equations:

$$\frac{T_i^{n+1} - T_i^n}{\Delta t} = \frac{\kappa}{\Delta x^2}(T_{i+1}^{n+1} - 2T_i^{n+1} + T_{i-1}^{n+1}) \quad \text{(forward Euler)} \tag{3}$$

$$\frac{T_i^{n+1} - T_i^n}{\Delta t} = \frac{\kappa}{\Delta x^2}(T_{i+1}^{n} - 2T_i^{n} + T_{i-1}^{n}) \quad \text{(backward Euler)} \tag{4}$$

Combining these results in the Crank-Nicolson scheme:

$$\frac{T_i^{n+1} - T_i^n}{\Delta t} = \frac{\kappa}{2\Delta x^2}\left((T_{i+1}^{n+1} - 2T_i^{n+1} + T_{i-1}^{n+1}) + (T_{i+1}^{n} - 2T_i^{n} + T_{i-1}^{n})\right) \tag{5}$$

Because of the implicit nature of the Crank-Nicolson scheme, an algebraic equation or linearizing the equation is necessary to solve the next time step. In our case, we can use the boundary conditions $T(x=0,t) = f(t)$ and $T(x=D,t) = 0$, where

$f(t)$ represents the arbitrary temperature forcing function (Fig. 2). Although the method is unconditionally numerically stable for the heat equation (Thomas, 2013), unwanted spurious oscillations can occur if the time steps are too long or the spatial resolution is too small. To avoid this, we use the following stability criterion:

$$\kappa \frac{dt}{dx^2} \leq \frac{1}{2} \tag{6}$$

Meeting this criterion (Eq. 6) for both tested values of $\kappa$ and all five forcing datasets (Fig. 2), the simulated temperatures are produced at five minute and two centimeter resolution with float-point precision. The resulting generated data (e.g. Fig. 3) provides an ideal reference from which temperatures can be sampled in space and time to replicate field measurements from 'well-behaved' portions of vertical temperature profiles within supraglacial debris, meaning subsets of the data that can be

shown to closely approximate a conductive system.

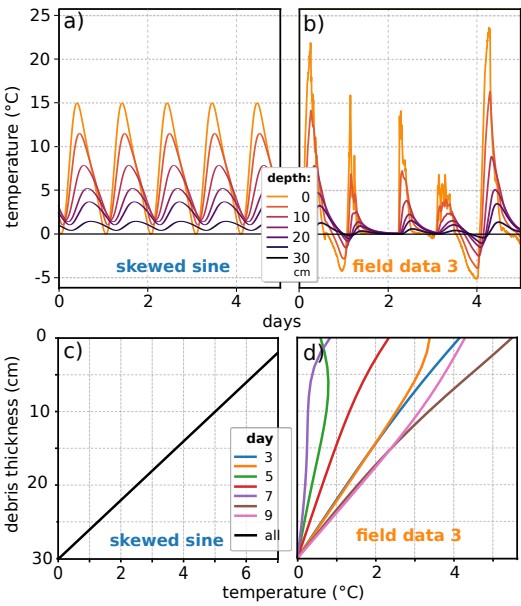

**Figure 3.** Five-day examples of the artificially generated debris layer temperature timeseries data for the skewed sine forcing (a) and the field data 3 forcing (b) for a 30 cm debris layer with $\kappa$ of $5 \cdot 10^{-7} m^2 s^{-1}$ using Crank-Nicolson scheme. (c and d) Daily averaged debris layer temperature profile for the full ten-day time series of the boundary conditions in the upper panels, show that the often-used steady-state assumption (Evatt et al., 2015) of the daily mean debris layer temperature, shown by a linear temperature gradient, is only fulfilled for periodic daily temperature forcings.

## 3.2 Experiments performed

We apply the Conway and Rasmussen (2000) method of deriving apparent $\kappa$ for a selected range of analytical set ups as described in the following subsections. When calculating $\kappa$ from data resampled from the synthetic cases, we calculate a single

diffusivity value for the last eight days of each forcing dataset, although the interactive tool also offers the option to calculate

$\kappa$ at a daily scale for assessment of field datasets. The calculation of the centered spatial derivatives is suitable for for unequal grid spacing, but we do not include analysis of unequal vertical thermistor spacings in this study as this was presented in a previous study (Laha et al., 2022). The properties of the analytical set up that are varied are: $\Delta t, \Delta x$, varying the precision of the temperature data, and adding Gaussian noise to assess statistical uncertainty. The performance of each experiment at recovering the known $\kappa$ prescribed in the artificial data, is assessed by calculating the relative error:

$$\text{Relative error} = \frac{\kappa_{\text{True}} - \kappa_{\text{Estimated}}}{\kappa_{\text{True}}} \tag{7}$$


Positive relative error values thus correspond to an underestimation of $\kappa$ compared to the known value. As effects of individual potential sources of error are contingent on other properties of the experimental set up, we present illustrative examples of the error tendencies and their co-dependencies over a range of properties. The full potential parameter space can be explored in the interactive tool. First the synthetic data is resampled without any added sensor or installation uncertainty to examine the

behaviour of numerical truncation errors. Subsequently the errors associated with the sensor and installation uncertainty are presented.

### 3.2.1 Quantifying truncation errors in space and time

In theory, the numerical solution to the diffusion problem should be equal to the analytical solution for infinitesimally small spatial and temporal sampling intervals. Truncation errors are expected to scale with the temporal and spatial increment of the

analysis with respect to the diurnal forcing cycle (Laha et al., 2022). Higher-order approximations would reduce the truncation error, but errors due to measurement uncertainties would dominate, as described by Zhang and Osterkamp (1995).

$$\lim_{\Delta t \to 0} \frac{T_{t+1} - T_{t-1}}{2\Delta t} = \dot{T} \qquad \& \qquad \lim_{\Delta x \to 0} \frac{T_{x+1} - T_x + T_{x-1}}{(\Delta x)^2} = T'' \tag{8}$$

For $\Delta x, \Delta t = 0$ the equations are not solvable.

For the temporal truncation error, we resample the artificial data both by skipping and by averaging over an increasing $\Delta t$

(Fig 4) from 5 minutes (the native resolution of the artificial data) to 6 hour intervals, to encompass the highest and lowest resolution temporal sampling of published field data (Appendix B). When skipping, we select every $n$-th value and omit the rest. When averaging, we take the mean temperature over $n$ values. While most studies store samples of the thermistor data at fixed $\Delta t$, we include an assessment of this averaging approach as some published field data collection campaigns are based on measurements of temperatures averaged over $\Delta t$ (e.g. Rowan et al., 2021). For the spatial truncation error, we resample by

skipping data points in space over a range of intervals to decrease the resolution of the 2 cm resolution artificially produced data. For this analysis we use the highest resolution temporal forcing with $\Delta t$ of 5 minutes and calculate $\kappa$ for the center of the debris layer, expanding $\Delta x$ symmetrically around this point. For assessing truncation errors due to both temporal and spatial resampling, the temperature values are used with their float point accuracy from the generated data, which implies perfect sensor precision.

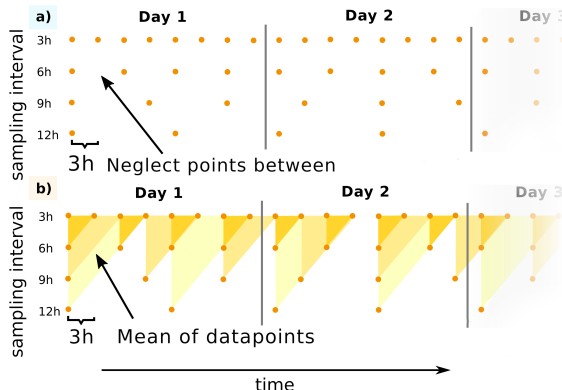

**Figure 4.** Illustrating the two different temporal resampling methods by displaying the temporal grid for different sampling intervals. We compare the method by skipping every $n$-th gridpoint (a - blue background) or by averaging over $n$ gridpoints (b - orange background).

### 3.2.2 Quantifying sensor and installation errors

Thermistors used to record supraglacial debris temperature profiles over time have varying manufacture stipulated sensor precision, and there may be uncertainty around their exact location in the debris cover as this can be challenging to measure with a high degree of accuracy in the field and can change if the debris moves.

To simulate the effect of temperature measurement precision, we discretize the temperature data to correspond with the measurement precision of $0.1$ to $0.4\,°\mathrm{C}$, which is representative of the precision of thermistors typically used in the field. The error properties of these differing sensor precisions are examined for a range of spatio-temporal resampling, in which we ensure symmetrical resampling of $\Delta x$ by resampling from the center of the debris layer outwards. Because the observed temperature changes and gradients are smaller at depth, it is expected that a higher precision of temperature measurement is required to capture them. Therefore we also examine how the relative error due to sensor precision varies with the depth in the debris layer at which the analysis is performed. For this we also consider the potential gain from even higher precision sensors by including a $0.01\,°\mathrm{C}$ temperature discretization, although this is more precise than any of the thermistor properties reported n the literature.

To simulate cases where either the vertical location of the temperature measurement is inaccurate, or the thermistor is displaced vertically over time, we use the sampled temperatures at float precision and add a time invariant vertical offset to each temperature measurement position. Each offset value is randomly sampled from a Gaussian distribution with standard deviation of $0.5$ centimeters around the true vertical measurement position, to represent an inaccurate field measurement of the vertical position. If thermistors move within the debris due to settling or debris migration, the positional inaccuracy could be even be larger, but this would likely be discernible from evidence of debris movement or identified when the thermistors were removed from the debris layer, allowing affected data to be excluded from further analysis. For both analyses of the effects of sensor precision and location accuracy we present only the idealised sinusoidal forcing data to best isolate the systematic error patterns and how they co-vary with the truncation errors established by the first analysis steps (Section 3.2.1).

### 3.3 Statistical uncertainty estimation

The method of Conway and Rasmussen (2000) is only valid in well-behaved conductive systems, and therefore the aim is to only apply the method to a time period and vertical section where this assumption is largely fulfilled. Therefore our error analysis so far assumes the debris to be a purely conductive, vertically and horizontally homogeneous system, while in nature, the debris cover will not be perfectly homogeneous and some non-conductive processes are expected to be contributing even in 'well-behaved' sections to temperature data.

To show that the model related studied error sources remain relevant despite additional external error sources we add random statistical noise to the data time-series that we perform our analysis on. For this we use the pure sine curve forcing for a $100 \, \text{cm}$ thick debris layer, with $\Delta x$ of $2 \, \text{cm}$ and $\Delta t$ of $5$ resolution for a $\kappa$ of $5 \cdot 10^{-7} m^2 s^{-1}$. Subsequently each individual float precision temperature value of the generated temperature time-series is modified by a value randomly sampled from a Gaussian distribution with a mean value of $0 \, °\text{C}$ and a standard deviation of $0.1 \, °\text{C}$. This procedure is repeated 20 times to generate a small ensemble of individually perturbed temperature time-series. The introduction of this statistical noise of $\sigma_T = 0.1 \, °\text{C}$ does not account for any specific physical processes, since non-conductive processes and effects due to spatial inhomogeneity would produce systematic temperature shifts on a multi-hourly to seasonal time-scale, as observed in some field data sets (Conway and Rasmussen, 2000; Nicholson and Benn, 2012; Petersen et al., 2022). The $\sigma_T = 0.1 \, °\text{C}$ is rather selected to statistically perturb the model system and simulate the effect of additional errors. By increasing or decreasing the selected $\sigma_T$ value the effect of the perturbation is respectively amplified or attenuated, but the general impact remains the same. The data is analyzed as in the previous sections by varying the temporal sampling interval and the vertical position in the debris layer for three selected vertical grid spacings $\Delta x$ ($4 \, \text{cm}$, $8 \, \text{cm}$, $16 \, \text{cm}$) to capture the co-dependencies of the error properties with these measurement choices. The temporal resampling is performed by skipping to preserve the maximum temperature perturbations to illustrate the effects of a maximum perturbation. When resampling by averaging, the perturbed values would equal out for longer temporal averaging periods. For each parameter combination the mean of $\kappa$ is calculated from the ensemble with a respective standard deviation to display the value spread.

## 4 Results

While the interactive tool provided allows a full range of sampling strategies to be explored, here we present results for selected cases within the range of realistic instrumental set ups. Our focus is to provide illustrative examples that characterize the error properties of each individual source.

### 4.1 Error due to temporal truncation

We illustrate the behaviour of the temporal truncation error calculated for a $100 \, \text{cm}$ thick debris layer with $\kappa$ of $5 \cdot 10^{-7} m^2 s^{-1}$, for up to 6 hours sampling intervals for both skipping and averaging resampling methods. As few field studies use $\Delta x$ as small

as our $2\,cm$ resolution artificial data, we show an example with $\Delta x$ of $6\,cm$ to better represent field observations. We show the behavior at two depths within the debris layer to illustrate the depth dependency of the error behavior.

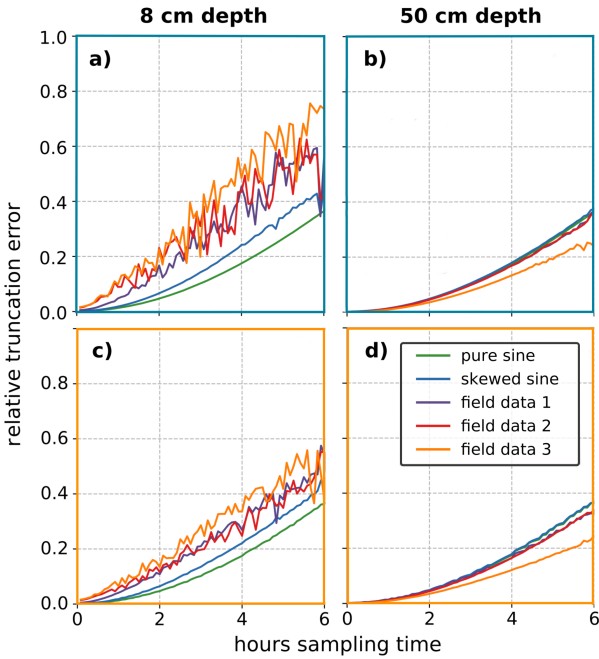

**Figure 5.** Relative temporal truncation error of recovering $\kappa$ using different temporal sampling intervals: Comparison of different temperature forcing for skipping (a,b - blue boundary) and averaging (c,d - orange boundary) re-sampling methods for two different depths in the $1.0\,m$ debris layer with a target $\kappa$ of $5 \cdot 10^{-7}\,m^2\,s^{-1}$.

The relative error in $\kappa$ due to temporal truncation error shows a general pattern of monotonic increase with increasing $\Delta t$
for the skipping method (Fig. 5a,b). Consistently positive relative errors indicate that increasing temporal sampling interval systematically underestimate $\kappa$. At shallow depths, the less sinusoidal the temperature forcing is, the larger the error at all sampling intervals (illustrated by the $8\,cm$ depth cases shown in Fig. 5a, c). A greater depths, the error for the sinusoidal forcing remains similar to that in the near surface, while the noisy surface diurnal signals are smoothed at depth and the associated error tends to be more similar to those of the sinusoidal surface forcing (illustrated by the $50\,cm$ depth cases shown
in Fig. 5b, d). When data is resampled by averaging, the temporal truncation error is very similar for the sine curve but, for the noisy field forcing data, averaging reduces the error compared to the skipping resampling method (Fig. 5c, d). These patterns of error behavior are also seen for $\kappa$ of $10 \cdot 10^{-7}\,m^2\,s^{-1}$.

Considering the maximum relative error produced by typical field installations, we can take the case of calculating diffusivity at a point as close to the surface as is reasonably possible at $4\,cm$, requiring a thermistor spacing of $2\,cm$, combined with the
longer typical time sampling interval of 1 hour, and calculating over a period with noisy surface forcing. This combination yields a maximum temporal truncation relative error of $25\%$. To minimize the error from a truncation perspective, a minimum

temporal resolution is desirable, and selecting days with surface temperature forcing that is closer to sinusoidal will decrease errors that may otherwise be significant at shallow depths.

## 4.2 Error due to spatial truncation

We illustrate the behaviour of the spatial truncation error calculated for a 100 cm thick debris layer with $\kappa$ of 5 and $10 \cdot 10^{-7}\, m^2\, s^{-1}$, for $\Delta x$ up to $50$ cm, using a sample of the five surface forcing datasets at $\Delta t$ of 5 minutes.

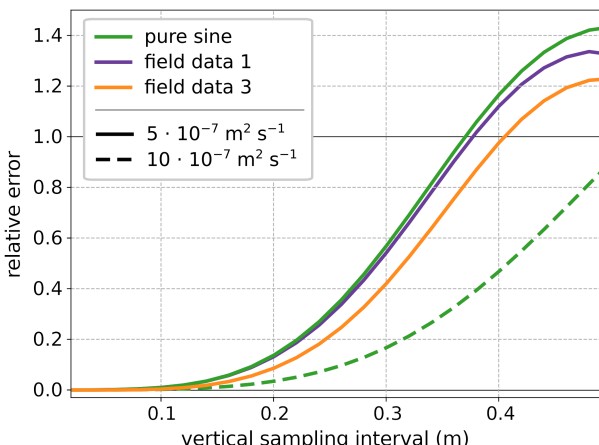

**Figure 6.** Comparison of the spatial truncation error for two different $\kappa$ values and forcing types, calculated for the central position in a $1.0$ m debris layer for symmetrically increasing $\Delta x$. For clarity we show only one curve for the higher diffusivity value, as all curves are shifted similarly when varying the target $\kappa$. The forcing datasets are at float precision with $\Delta t$ of 5 minutes.

Spatial truncation error values (Fig. 6) remain quasi-constant for low $\Delta x$, up to when the centered differencing scheme spans more than $20$ cm, and thereafter increase rapidly with increasing $\Delta x$. The spatial truncation error is relatively insensitive to the different surface temperature forcings, and in contrast to the temporal truncation error does not vary markedly with debris

depth. Instead the $\kappa$ imposes a strong influence, with higher $\kappa$ having smaller errors, shifting the respective curves to the right as shown for the case of the sinusoidal forcing in Fig. 6. Given that the diffusivity is the target of sensor installations, this parameter cannot be known in advance and the results suggest that $\Delta x$ of below 14 cm is desirable to minimise spatial truncation errors across a range of potential $\kappa$. The consistently positive error values mean that the spatial source of truncation error also has the tendency to systematically underestimate $\kappa$, and increasingly so with more widely spaced temperature measurements.

## 270 4.3 Error due to thermistor precision

To illustrate the role of temperature sensor precision, we first focus on the range of sensor spacings that are not affected by the spatial truncation error, i.e. for $\Delta x$ up to $14$ cm (Fig. 7), and show the relative error for a $\Delta t$ ranging from 5 minutes to several hours. The error due to temperature discretization is generally less pronounced for smaller temperature discretizations, representing greater thermistor precision. Maximum errors occur for small values of $\Delta x$, decreasing to stable relative errors of

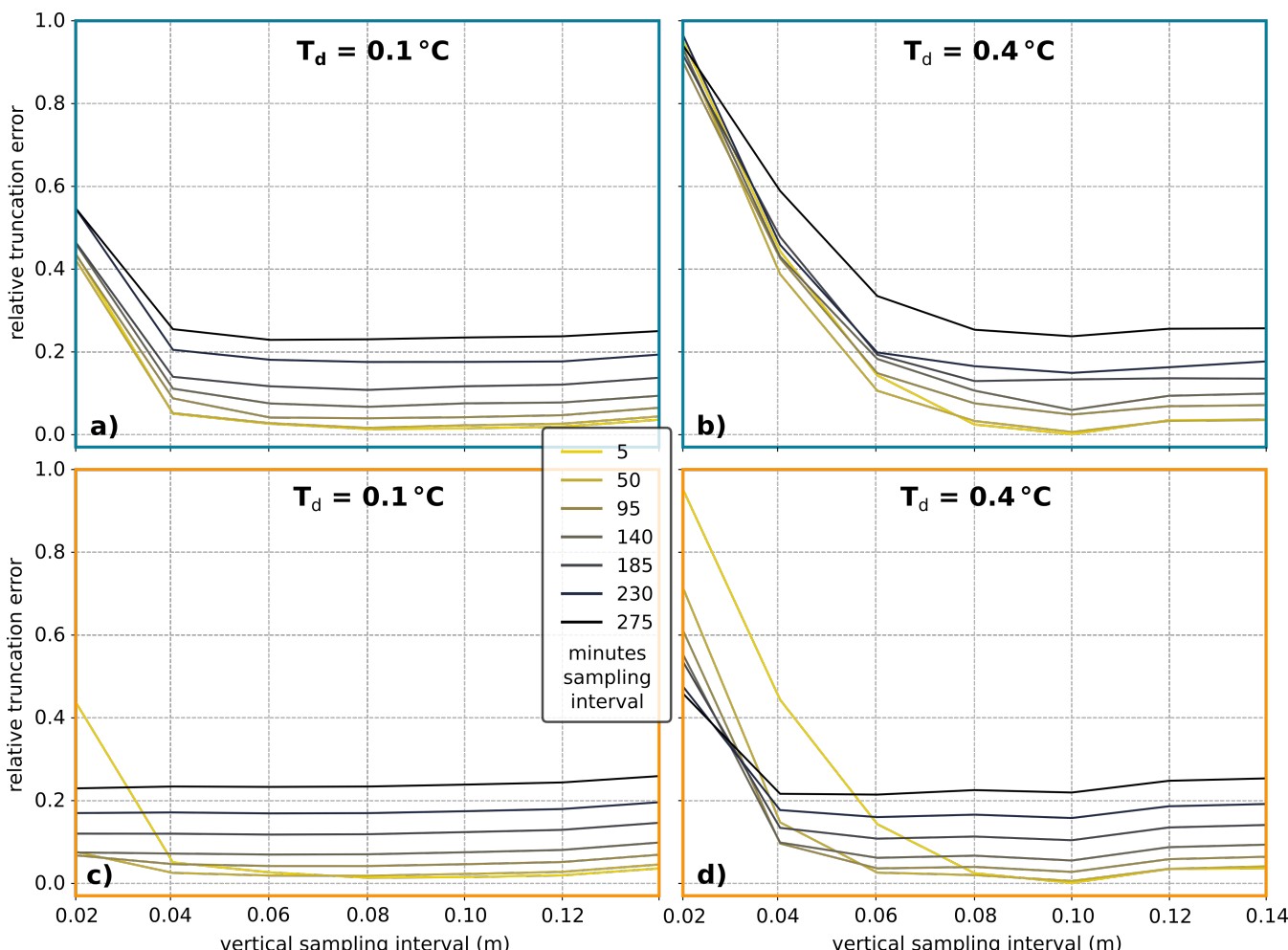

**Figure 7.** Relative error of estimated $\kappa$ due to thermistor temperature discretisation of $0.1$°C and $0.4$°C for vertical sampling intervals up to $0.14$ m and $\Delta t$ resampled by skipping (a, b) and averaging (c,d)) for the intervals shown in the legend, such that the 5 minute dataset is identical for both methods. The case presented is a centered sampling of a $0.3$ m thick layer with target $\kappa$ of $5 \cdot 10^{-7} \, m^2 \, s^{-1}$ forced with a sinusoidal surface forcing.

$< 25\%$ for $\Delta x > 6$ cm, above which the error also decreases systematically with decreasing temporal sampling interval. Values of $\Delta x$ between the dominant spatial truncation error (Section 4.2) and the error due to the sensor precision are desirable, so between ca. 6 and 14 cm for the representative parameter space explored in our analyses.

The depth dependency of the error associated with discretization indicates the importance of high precision sensors for sampling the debris at depth (Fig. 8). For a $\Delta x$ of $2$ cm, only measurements with a maximum thermistor uncertainty of $0.01$ °C would produce correct values, and then only for the first $20$ cm of debris. Increasing $\Delta x$ to $6$ cm, the relative error decreases for all curves. Still, the thermistors used in most field experiments, which have reported precision ranging from $0.1$ to $0.4$ °C would

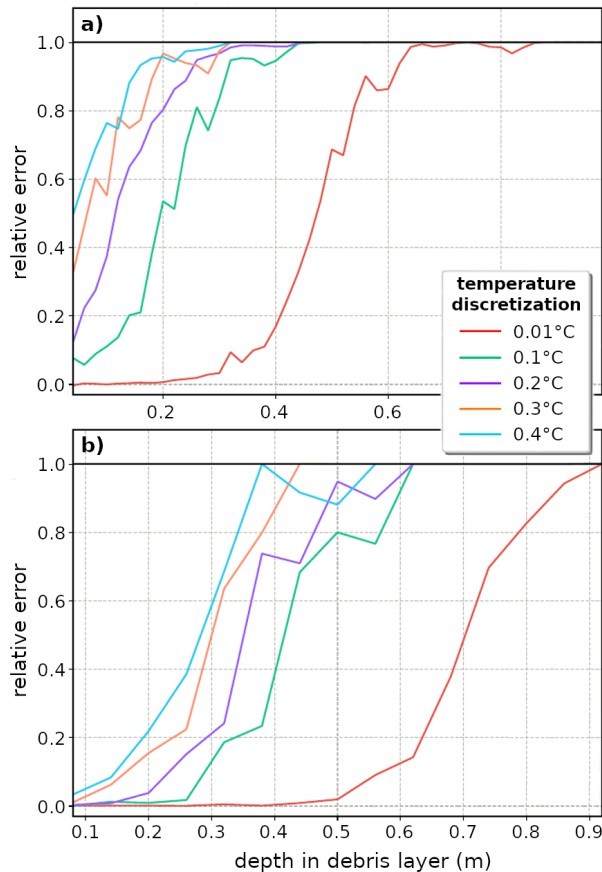

**Figure 8.** Relative error of estimated $\kappa$ due to thermistor discretization by depth for a $1.0\,\text{m}$ debris layer, with 5 minute sinusoidal surface forcing for $\Delta x$ of (a) $2\,\text{cm}$ and (b) $6\,\text{cm}$, with the different color lines corresponding to different values of temperature discretization.

not produce correct values at depth. For the case shown it would become difficult to obtain reliable values at depths beyond $60\,\text{cm}$ even with high-precision thermistors. The error behavior is dependent on capturing temperature gradients sufficiently well, so the specific error limits are dependent on the amplitude of the surface forcing fluctuations and the diffusivity as well
as the chosen discretization and spatio-temporal sampling. For a given discretization, meaningful values can be obtained at greater depth by enlarging the $\Delta x$, but higher precision sensors are always an advantage. As for both types of truncation error, the sensor precision error systematically underestimates the target $\kappa$.

### 4.4  Error due to vertical thermistor position inaccuracy

Conway and Rasmussen (2000) report that a vertical error of $0.5\,\text{cm}$ would result in a marginal temperature difference of $0.1\,°\text{C}$
and $0.02\,°\text{C}$ for their measurement setups. They and others (e.g. Nicholson and Benn, 2012) interpret this to mean that a vertical thermistor displacement would not affect the results as long as this value does not change in time.

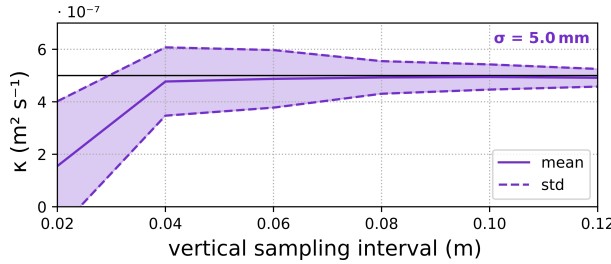

**Figure 9.** Illustrating the influence of thermistor displacement on estimated $\kappa$, by randomly displacing locations by a normal distribution with standard deviation of $0.5\,\mathrm{cm}$ over a range of vertical spacing intervals. The true/target thermal diffusivity is shown by the horizontal black line, showing that for small temperature sampling intervals sensor displacement results in large inaccuracies in $\kappa$.

Our analysis, however, shows that low accuracy knowledge of the temperature measurement location could produce a systematic error for smaller $\Delta x$. For example, in the relatively rare case that sensors are installed with a $\Delta x$ of two centimeters, the resultant error on calculated values of effective thermal diffusivity is so large that the data would become unusable. With increasing $\Delta x$, the relative error decreases, such that the mean $\kappa$ over the depth of the layer recovers the target value. This error source is the only one of this study that has the potential to increase $\kappa$ values, as shown in Fig. 9 by the spread of $\kappa$ above the known reference value.

### 4.5 Statistical uncertainty estimation

In contrast to the noise-free case shown in Figure 5, with the addition of statistical noise, the relative temporal truncation error now increases with depth and the predominance of relative errors $< 25\%$ in the sub-hourly $\Delta t$ range can now only be recovered in the near-surface portion of the debris layer. The standard deviation of the error curves nearer the surface is less than a few percent of the relative error therefore showing a minimal ensemble spread, while at depth the ensemble spread is larger. From this we can see that where the random noise introduced is large compared to the spatio-temporal temperature gradients, as is the case at greater depths in the debris layer, the method essentially is no longer applicable. Increasing the $\Delta x$ decreases the relative error found at depth but has little impact on the smaller errors nearer the surface. At larger $\Delta x$ even the near surface values now have non-zero relative error for short $\Delta t$; this is due to the spatial truncation error of the vertical sampling interval as displayed in Fig. 6 coming into play, while at greater depth in the debris the larger $\Delta x$ decreases the relative error although this still remains $> 0.6$ with a large relative error standard deviation values of $\sim 10\%$. In our example, the combination with the most precise recovery of the target $\kappa$, with relative error approaching zero, was for $\Delta x = 8\,\mathrm{cm}$ at a $18\,\mathrm{cm}$ depth and at a $5$ min. temporal sampling interval. For this combination the relative error due to temporal truncation error increases to $\sim 10\%$ and $\sim 20\%$ at $\Delta t$ of $120$ min. and $240$ min. respectively.

Displaying the noise-induced relative error of $\kappa$ more explicitly in relation to the depth in the debris over the span of the shared calculation range $(0.18 - 0.82\,\mathrm{cm})$, highlights that there are characteristic transition zones between where the method is still applicable and where it is not and, as these scale with the relative magnitude of the noise, the transition location is dependent

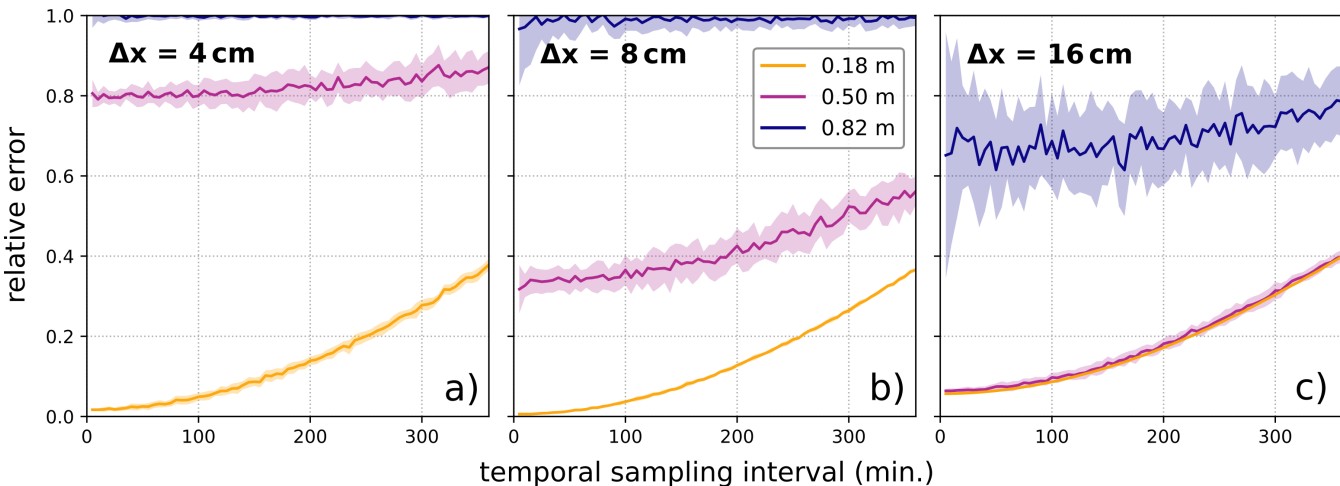

**Figure 10.** Relative errors of thermal diffusivity of statistically perturbed ensemble data for a $1.0\,\text{m}$ debris layer varied by temporal sampling interval for three different depths in the debris layer and three different $\Delta x$ ($4\,\text{cm}$, $8\,\text{cm}$, $16\,\text{cm}$). The ensemble consists of 20 cases with each individual temperature value being perturbed by a Gaussian distribution with a standard deviation of $\sigma_T = 0.1\,^\circ\text{C}$. The solid line is the mean relative error value and the shaded background represents the standard deviation of the relative error.

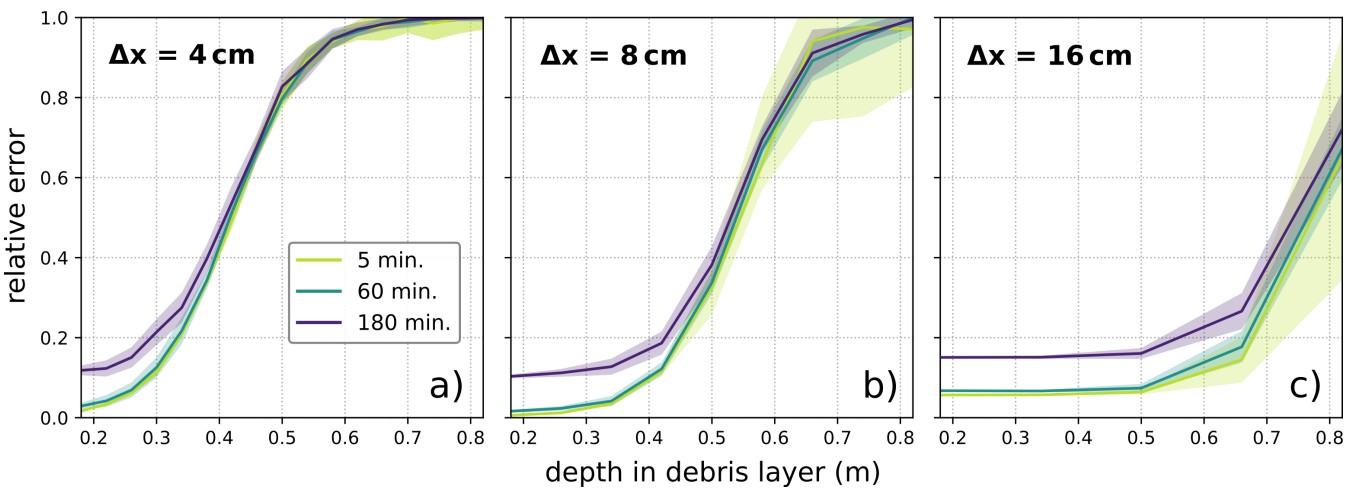

**Figure 11.** Relative errors of thermal diffusivity of statistically perturbed ensemble data for a $1.0\,\text{m}$ debris layer varied by the depth in the debris layer for three different sampling intervals and vertical grid-spaces $\Delta x$ ($4\,\text{cm}$, $8\,\text{cm}$, $16\,\text{cm}$). The ensemble consists of 20 individual runs of each 8 days with each individual temperature value being perturbed by a Gaussian distribution with a standard deviation of $\sigma_T = 0.1\,^\circ\text{C}$. The solid line is the mean relative error value and the shadowed background represents the standard deviation of the relative error.

315   on the $\Delta x$ used in the analysis (Fig. 11), as well as the amplitude of the surface forcing, the diffusivity and the temperature discretization. For example, for the upper-most section of the artificial debris layer all curves with $5\,\text{min.}$ and $60\,\text{min.}$ sampling intervals provides relative errors below $10\,\%$, while in the data combination we show, the transition to relative error $> 20\%$

for these sampling intervals is $0.3\,\mathrm{m}$, $0.45\,\mathrm{m}$, $0.65\,\mathrm{m}$ depth for vertical grid-spacings of $4\,\mathrm{cm}$, $8\,\mathrm{cm}$ and $16\,\mathrm{cm}$ respectively. Therefore, as is the case for the depth dependence of temperature discretization (Fig. 8), increasing the $\Delta x$ increases the depth at which meaningful values to be recovered when noise is present. However the increasing grid-spacing also results in a $\Delta x$ truncation error which is visible in Fig. 11c as a vertical displacement in relative error values additional to the displacement caused by the temporal truncation error.

## 5 Discussion

In previously published values, most apparent thermal diffusivity derived using the method of Conway and Rasmussen (2000) are below $10 \cdot 10^{-7}\,m^2\,s^{-1}$, typically ranging from 1 to 30 with some outlier values exceeding $100 \cdot 10^{-7}\,m^2\,s^{-1}$ (see Table 2 in Laha et al. (2022)). The implementation errors that our analysis reveals are often comparable to this range of published values, highlighting how relevant it is to correctly consider the numerical errors in choosing how to apply this method.

While the interactive tool accompanying our analysis allows a wider range of the parameter space to be explored, the cases we present were chosen to characterize the main numerical error sources inherent in the method within the parameter space of published values (B). The numerical and measurement implementation error sources investigated here all tend to systematically underestimate $\kappa$, while the relative error associated with uneven thermistor spacing (tested for a 3 thermistor cases by Laha et al. (2022)) was previously identified to systematically overestimate $\kappa$ by up to $50\,\%$ at thermistor spacing rations of 1:5.

In general the numerical errors associated with applying this method are all related to how well the temperature gradients in space and time within the debris cover can be captured by the instrumental set up. Temporal truncation errors in the absence of statistical noise are typically $< 25\%$ in most expected deployment settings, at sampling intervals of $\leq 60$ minutes. Near-surface measurements suffer more error because the diurnal temperature cycle at the surface is most non-sinusoidal and therefore produces larger temporal truncation errors. Consequently, conditions that more closely approximate sinusoidal conditions (i.e clear sky stable atmospheric conditions) reduce the errors in the near surface layers, but this becomes less relevant at depth, as surface noise introduced by weather is progressively smoothed out at greater depth in the debris. Spatial truncation due to the choice of thermistor spacing is not very sensitive to the non-sinusoidal forcing, but becomes $\geq 25\%$ at $\Delta x$ above $\sim 25\,\mathrm{cm}$ for the range of $\kappa$ reported in the literature, and the error is larger for smaller $\kappa$. The $\Delta x$ range at which errors are small and similar regardless of the forcing and $\kappa$ is $\leq 14\,cm$, providing a conservative upper bound to limit spatial truncation errors. Even though a $\Delta t$ or $\Delta x \longrightarrow 0$ would produce a minimal truncation error, too small sampling intervals also can produce erroneous results because for a $\Delta t \longrightarrow 0$, the linear regressions coefficient of determination decreases strongly. In practice, this is not a problem for the temporal sampling since short temporal sampling intervals can always be resampled afterwards. A more significant problem occurs if low precision thermistors are positioned too close to each other, especially if the profile is comprised of only a few thermistors, making it impossible to spatially resample the temperature data. While this effect diminishes to a stable value of relative error $\geq 25\%$ for $\Delta x$ above $\sim 6\,cm$, with increasing depth, the thermistors must be further apart otherwise the thermistor measurement uncertainty dominates the measurement. Therefore, although the highest precision thermistors should always be chosen if possible, using thermistors with maximum precision becomes even more important at greater depths in

the debris layer. The only error source investigated here that has the potential to overestimate $\kappa$ is that due to inaccurate temperature measurement location. This can happen due to poorly measured positions, or due to debris settling after sensor installation of if the thermistor profiles are installed on a slope, that is subject to gradual gravitational sliding or reworking. In contrast to Conway and Rasmussen (2000), who showed that a constant error in the thermistor position was not important to the analysis, we find that, at least for very small $\Delta x$ the calculated $\kappa$ does depend on the thermistor positions relative to each other being correctly known and sustained over the measurement period. However, thermistors are typically placed more a than a few centimeters apart this error source might be expected to have little effect if the $\kappa$ is calculated at several levels int he debris cover, as the mean value of the location perturbed cases recover the target diffusivity. Introducing a statistical noise term highlights the manner in which noise degrades the temperature gradients that the method relies on, particularly at greater depths in the debris where temperature variations in space and time are small compared to the introduced noise term. Thus care must also be taken to assess if the method is being applied to portions of the debris layer where the gradients are well captured.

In the best practice guidelines (Appendix A) we address all sources of methodological error discussed in this paper suggesting optimal implementation strategies for future field studies that wish to deploy these methods of analysing representative thermal conductivity of natural debris layers following the method of Conway and Rasmussen (2000). Our recommendations differ somewhat from those of Laha et al. (2022), as the purpose is different. While Laha et al. (2022), sought to determine the optimal method to determine sub-debris ablation rates directly from temperature sensors, using a minimal number of thermistors, we seek to understand the best way to determine a representative $\kappa$, from which effective thermal conductivity suitable for onward use in generalized surface energy balance models can be derived. For their purpose, they propose to "set the sensor spacing to be $1/5^{\text{th}}$ of the debris thickness at the location", however the non-linear nature of the single error sources presented in this paper indicates that we cannot generalize such statements if the goal is parameter determination, rather than direct ablation determination. Furthermore, they stated "the top sensor should be placed approximately at the middle of the debris layer", as this captures the relevant flux being delivered to the underlying ice. Our analysis indicates that while it is true that thermistors too close to the surface produce large truncation errors, the same is valid for too deep thermistors as the temperature gradient is to small relative to the thermistor precision. By providing an open source interactive tool that can be used to explore all the methodological sources of error in implementing the most widely-used method of determining $\kappa$ we offer a ready-to-use means for determine the field set up that minimises these numerical methodological errors. The intention is that, prior to a new field deployment, the error response of the expected conditions of debris thickness, surface forcing amplitude, sensor number and precision can be explored and the best possible field deployment of sensors can be made.

In addition to the errors related to measurement set up and analysis procedure investigated in this study, non-conductive processes within the debris layer (e.g. rain, phase changes) can also be present(Conway and Rasmussen, 2000; Nicholson and Benn, 2012; Petersen et al., 2022). Unfortunately, it is not always clear in the published literature that the thermal diffusivities and associated thermal conductivity values were derived from optimal conditions sampled within the dataset. The suitability of the sampled debris temperature profiles for determining debris thermal parameters must be carefully evaluated on a case-by-case basis using meteorological data and closely evaluating the measurements and their gradient functions (Petersen et al., 2022) in order to establish that the data sub-set represents predominantly conductive conditions, before applying the method

of Conway and Rasmussen (2000). Once a suitable effective thermal conductivity is established based on 'well-behaved' conditions these base values can be modified in implementation within a surface energy balance model to account for changes in the pore fluid type to allow simulation of varying wet/dry debris conditions (Collier et al., 2014; Giese et al., 2020).

The recently published database of supraglacial debris properties, DebDab v1 (Fontrodona-Bach et al., 2025), reveals that from the 176 values of debris thermal conductivity, only 33 report an associated uncertainty, and while 121 include the debris layer thickness, only 23 report on details such as the thermistor depth. To facilitate intercomparison of these data it would be valuable to include the temporal sampling used, as well as the rock properties and porosity used to convert $\kappa$ to thermal conductivity. Deeper consideration and potential common reanalyses of these data would require the original thermistor data to be publicly available, which is not always the case. Reanalyzing previously published vertical temperature profiles with common resampling strategies, based on the findings of this study, would facilitate intercomparison of $\kappa$ values, while reanalysis using the methods of Petersen et al. (2022) and/or Laha et al. (2022) might yield more robust and representative global values by providing respectively a more rigorous assessment of non-conductive processes and inclusion of multilayered thermal properties within the natural debris layers that have been sampled.

## 6 Conclusion

Conway and Rasmussen (2000) provide a practical method to estimate thermal diffusivity values from a vertical array of thermistors in supraglacial debris layer, which is applicable for spatially homogenous debris which is behaving as a close approximation to a purely conductive system. Although this method has become the standard method for determining effective thermal conductivity to be used in surface energy balance models of sub-debris ice ablation (e.g. Nicholson and Benn, 2006, 2012; Juen et al., 2013; Rounce et al., 2015a; Chand and Kayastha, 2018; Rowan et al., 2021), our analysis demonstrates several ways in which the derived $\kappa$ is sensitive to numerical errors related to instrumental set up and analysis choices, even when solving for a pure conduction case. The method has regularly been used without considering these error sources, making it difficult to robustly compare published values derived using this method.

To address this we provide an open source tool (https://github.com/calvinbeck/TC-DTD) where researchers can investigate the combined opportunities and limitations of applying the method by Conway and Rasmussen (2000) to glaciology and beyond. We hope this facilitates more consistent and rigorous experimental design in future field measurements determining debris thermal properties, by allowing users to simulate their own artificial data, that most closely approximates their planned field site, and repeat all our analyses presented here with their own artificial or field datasets.

In this paper we used this tool to provide illustrative examples of the magnitude and tendencies of the systematic errors associated with individual instrumental and analytical choices. Based upon our findings we provide a set of best-practice guidelines (A) to minimize systematic errors in applying the method of Conway and Rasmussen (2000). While recent publications highlight limitations of the simplest deployment of the heat diffusion equation in natural debris layers due to the role of non-conductive processes and internal debris stratification (Laha et al., 2022; Petersen et al., 2022), our analysis and best practice guidelines show the sampling strategies that will yield the best results, provided that the temperatures underpinning

the analyses demonstrably sample conditions that closely approximate a homogeneous conductive system. Our analysis also highlights that it is challenging to interpret derived debris thermal properties if the sensor and analysis system is not reported and accounted for. In the light of this we encourage more rigorous reporting of implementation strategies and uncertainty in order to facilitate cross-comparison of reported results.

*Code availability.* Publicly available under: https://github.com/calvinbeck/TC-DTD

*Author contributions.* This publication is based on the MSc thesis of CB, supervised by LN. LN conceived the study, CB performed all analysis, developed the interactive tool, produced the figures and led the preparation of the manuscript. Both CB and LN worked to finalise the manuscript for publication.

*Competing interests.* The authors declare that they have no conflict of interest.

*Acknowledgements.* Field datasets used for temperature forcing in this analysis (Fig. 2) were provided by Mohan Chand, Rijan Kayastha, Martin Juen and Christoph Mayer. In the course of the Masters thesis analysis further forcing data was provided by members of the IACS working group on Debris Covered Glaciers (https://cryosphericsciences.org/activities/wgdebris/).

## Appendix A:  Best practice guidelines

Our analysis leads us to the following best practice guidelines to help other researchers to get as much as possible out of their measurements.

> **Thermistor precision:**
> As small as possible, but not larger than 0.1 K.

**Debris layer thickness:**

To determine a representative thermal diffusivity from which robust, generally applicable, thermal conductivity values can be derived, sampling a minimum of $40\,cm$ but ideally deeper (e.g. 100 cm) debris thickness is advised. The maximum depth that can be meaningfully sampled is limited by the thermistor precision and temperature gradients in the debris layer, which can be simulated beforehand using the provided tool.

**Number of thermistors:**

The method requires at least three thermistors, but more thermistors make it possible to calculate diffusivity values for different depths and therefore makes it possible to identify non-conductive processes or other inconsistencies within the debris layer. With only three temperature sensors it is difficult to assess if the sampled debris meets the requirement of closely approximating a conductive system. A second redundant set of thermistors can also be helpful to rule out measurement errors.

**Thermistor installaton:**

Choose a site that is not expected to be subject to gravitational reworking or sliding of the debris, and where lateral heat fluxes are expected to be minimal. Place thermistors at equal vertical intervals of $8$ to $20\,cm$. Even though the uppermost layer often does not produce ideal results, it can be helpful to place a thermistor at or near the debris-surface to provide surface forcing data. Depending on the depth, the thermal diffusivity, and temperature gradient of the debris layer, the method produces more significant errors with a greater depth limiting the depth where it makes sense to place thermistors. The sweet spot can be determined by simulating the debris layer of interest beforehand with model parameters from previous measurements or other estimations.

**Thermistor recovery:**

Thermistors have to be carefully extracted, and their vertical positions recorded, at the end of the measurement period to make sure the thermistors haven't moved in the debris while deployed. In case the thermistors moved, it might be necessary to discard the dataset. Therefore mounting thermistors to a thermally insulated rod or set of rods so that their positions are fixed is a valuable approach to eliminate this potential error source.

440

**Temporal sampling interval:**

Sample with a temporal resolution as short as possible and then average over a 5 minute period. Over such a short period, the temperature is assumed to be nearly constant and therefore not to reduce gradients. By averaging the temperature over a short interval, discretization is reduced.

> **Measurement duration and conditions:**
>
> It depends on the scientific objective and seasonality, but at least a week of suitable stable meteorological conditions are needed. Therefore, if one has unlucky conditions, a measurement duration of several months could be necessary. A shorter period of predominantly sinusoidal surface temperature forcing and evidence that non-conductive processes are minimal is the best way to obtain robust values, so avoiding periods of precipitation, seasonal change and phase change is advised.

**Appendix B: Field measurement overview**

| Site | Glacier | Year | DT | SR | AC | TM | Thermistor | Start | Days |
|---|---|---|---|---|---|---|---|---|---|
| ID | | | m | min | °C | # | m | Date | # |
| KH1a | Khumbu | 2014 | 2.8 | 30 | ±0.4 | 8 | 0.1, 0.25, 0.4, 0.55, 0.7, 0.8, 0.9, 1.0 | 2014-05-10 | 188 |
| KH1b | Khumbu | 2015 | 2.8 | 30 | ±0.4 | 8 | 0.1, 0.25, 0.4, 0.55, 0.7, 0.8, 0.9, 1.0 | 2014-11-21 | 328 |
| KH2a | Khumbu | 2014 | 0.7 | 30 | ±0.4 | 8 | 0.0, 0.1, 0.2, 0.3, 0.4, 0.5, 0.6, 0.7 | 2014-05-13 | 184 |
| KH2b | Khumbu | 2015 | 0.8 | 30 | ±0.4 | 9 | 0.0, 0.1, 0.2, 0.3, 0.4, 0.5, 0.6, 0.7, 0.8 | 2015-10-20 | 338 |
| KH4 | Khumbu | 2014 | 0.3 | 30 | ±0.4 | 4 | 0.02, 0.11, 0.22, 0.3 | 2014-05-20 | 180 |
| KH5 | Khumbu | 2015 | 0.7 | 30 | ±0.4 | 8 | 0.0, 0.0, 0.2, 0.3, 0.4, 0.5, 0.6, 0.7 | 2015-10-20 | 205 |
| CN1 | Changri Nup | 2016 | 0.1 | 30 | ±0.1 | 3 | 0.01, 0.05, 0.1 | 2015-11-28 | 450 |
| CN2 | Changri Nup | 2016 | 0.08 | 30 | ±0.1 | 2 | 0.01, 0.08 | 2015-11-28 | 450 |
| CNW1 | Changri N. (W.) | 2010 | 0.1 | 30 | ±0.1 | 4 | 0.025, 0.05, 0.075, 0.1 | 2010-10-31 | 698 |
| CNW2 | Changri N. (W.) | 2012 | 0.125 | 30 | ±0.1 | 4 | 0.05, 0.075, 0.1, 0.125 | 2012-12-05 | 723 |
| CNW3a | Changri N. (W.) | 2014 | 0.21 | 30 | ±0.1 | 3 | 0.01, 0.16, 0.21 | 2014-11-30 | 309 |
| CNW3b | Changri N. (W.) | 2015 | 0.26 | 30 | ±0.1 | 3 | 0.02, 0.2, 0.26 | 2015-11-27 | 33 |
| CNW3c | Changri N. (W.) | 2017 | 0.1 | 30 | ±0.1 | 3 | 0.01, 0.05, 0.1 | 2017-11-26 | 347 |
| CNW3d | Changri N. (W.) | 2018 | 0.14 | 30 | ±0.1 | 3 | 0.02, 0.1, 0.14 | 2018-11-11 | 379 |
| NG1 | Ngozumpa | 2002 | 2.2 | 30 | ±0.1 | 6 | 0.0, 0.22, 0.33, 0.45, 0.65, 0.77 | 2001-11-13 | 323 |
| NG2 | Ngozumpa | 2015 | 2.0 | 360 | ±0.25 | 11 | 0.01, 0.2, 0.4, 0.6, 0.8, 1, 1.2, 1.4, 1.6, 1.8, 2 | 2014-12-06 | 484 |
| IM4 | Imja-Lhotse S. | 2014 | 1.6 | 30 | ±0.3 | 5 | 0.0, 0.1, 0.2, 0.4, 0.83 | 2014-05-31 | 162 |
| IM11 | Imja-Lhotse S. | 2014 | 0.45 | 30 | ±0.3 | 5 | 0.0, 0.05, 0.1, 0.2, 0.36 | 2014-05-31 | 162 |
| IM13 | Imja-Lhotse S. | 2014 | 0.33 | 30 | ±0.3 | 4 | 0.0, 0.05, 0.1, 0.2 | 2014-05-31 | 162 |
| IM14 | Imja-Lhotse S. | 2014 | 0.26 | 30 | ±0.3 | 3 | 0.0, 0.05, 0.24 | 2014-05-31 | 162 |
| ILS1 | Imja-Lhotse S. | 2013 | 0.3 | 30 | ±0.3 | 6 | 0.0, 0.05, 0.1, 0.15, 0.2, 0.3 | 2013-09-14 | 11 |
| ILS2 | Imja-Lhotse S. | 2013 | 0.47 | 30 | ±0.3 | 7 | 0.0, 0.05, 0.1, 0.15, 0.2, 0.3, 0.47 | 2013-09-14 | 11 |
| ILS3 | Imja-Lhotse S. | 2013 | 0.36 | 30 | ±0.3 | 6 | 0.0, 0.05, 0.1, 0.15, 0.2, 0.36 | 2013-09-14 | 11 |
| ILS4 | Imja-Lhotse S. | 2013 | 0.4 | 30 | ±0.3 | 7 | 0.0, 0.05, 0.1, 0.15, 0.2, 0.3, 0.4 | 2013-09-14 | 11 |

Table B1: Overview table (1/2) of thermal diffusivity field measurement sites. DT: Debris layer thickness; SR: Sampling rate; AC: Thermistor accuracy; TM: Number of thermistors; Data from Nicholson and Benn (2012); Rounce et al. (2015b); Rowan et al. (2021).

| Site ID | Glacier | Year | DT m | SR min | AC °C | TM # | Thermistor m | Start Date | Days # |
|---|---|---|---|---|---|---|---|---|---|
| LG1a | Lirung | 2013 | 0.38 | 5 | ±0.1 | 3 | 0.1, 0.2, 0.3 | 2013-09-24 | 9 |
| LG1b | Lirung | 2013 | 0.4 | 5 | ±0.1 | 3 | 0.01, 0.1, 0.4 | 2013-12-05 | 7 |
| LG1c | Lirung | 2014 | 0.4 | 5 | ±0.1 | 3 | 0.01, 0.1, 0.4 | 2014-04-06 | 13 |
| LG2a | Lirung | 2013 | 0.42 | 5 | ±0.1 | 3 | 0.05, 0.15, 0.35 | 2013-09-20 | 13 |
| LG2b | Lirung | 2013 | 0.4 | 5 | ±0.1 | 3 | 0.01, 0.1, 0.4 | 2013-12-05 | 7 |
| LG2c | Lirung | 2014 | 0.4 | 5 | ±0.1 | 3 | 0.01, 0.1, 0.4 | 2014-04-07 | 12 |
| SDF1 | Suldenferner | 2014 | 0.6 | 30 | ±0.3 | 3 | 0.0, 0.02, 0.06 | 2014-07-30 | 54 |
| SDF2 | Suldenferner | 2014 | 0.12 | 60 | ±0.3 | 5 | 0.0, 0.03, 0.06, 0.09, 0.12 | 2014-09-26 | 319 |
| SDF3 | Suldenferner | 2014 | 0.24 | 60 | ±0.3 | 6 | 0.04, 0.08, 0.12, 0.16, 0.20, 0.24 | 2014-09-26 | 319 |
| SDF4 | Suldenferner | 2014 | 1 | 60 | ±0.3 | 6 | 0.0, 0.2, 0.4, 0.6, 0.8, 1.0 | 2016-09-25 | 278 |
| BG | Belvedere | 2003 | 0.27 | 15 | ±0.3 | 4 | 0.04, 0.11, 0.23, 0.27 | 2003-06-24 | 42 |
| LB_dry | Larsbreen | 2002 | 0.65 | 10 | ±0.4 | 8 | 0.0, 0.09, 0.19, 0.29, 0.38, 0.53, 0.61, 0.75 | 2002-07-21 | 5 |
| LB_exp | Larsbreen | 2002 | 0.65 | 10 | ±0.4 | 5 | 0.1, 0.2, 0.3, 0.4, 0.5 | 2002-07-09 | 12 |
| LB_sat | Larsbreen | 2002 | 0.65 | 10 | ±0.4 | 8 | 0.0, 0.05, 0.1, 0.15, 0.2, 0.3, 0.35, 0.4 | 2002-07-03 | 6 |
| VF1 | Vernagtferner | 2010 | 0.08 | 5 | ±0.35 | 3 | 0.04, 0.06, 0.08 | 2010-06-24 | 83 |
| VF2 | Vernagtferner | 2010 | 0.18 | 5 | ±0.35 | 3 | 0.07, 0.11, 0.15 | 2010-06-24 | 82 |

Table B2: Overview table (2/2) of thermal diffusivity field measurement sites. DT: Debris layer thickness; SR: Sampling rate; AC: Thermistor accuracy; TM: Number of thermistors; Data from Nicholson (2005); Juen et al. (2013); Chand and Kayastha (2018); Rowan et al. (2021).

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
