# Peer review of "Numerical study of the error sources in the experimental estimation of thermal diffusivity: an application to debris-covered glaciers"

_EGUsphere, 2023_

## Referee Comment (RC2)

[referee-annotated manuscript omitted]

---

## Referee Comment (RC3)

[referee-annotated manuscript omitted]

---

## Author Comment (AC1)

**Review 1**

The authors analyse a commonly used finite-difference method for estimating thermal diffusivity of supraglacial debris using vertical debris-temperature profile. The aim of the study is to understand the effects of temporal and spatial discretisation on the uncertainty in estimated diffusivity. Due to the importance of debris thermal properties in computing the dynamics of debris-covered glaciers, this topic is of importance. However, the study appears to  have serious weaknesses in terms of the experimental design and the underlying assumptions. unless these issues are addressed with suitable (and doable) modification of the methods, the results and conclusions will remain weak.

We thank Dr Banerjee for his thoughtful and critical review of our work. In the light of this and the other reviewers comments we have substantially revised the manuscript. In particular we have:

- restructured and rewritten it to make our purpose and scope more clear.
- emphasised the utility of analysing this method, despite the developments of new approaches in e.g. Laha et al, 2022.
- clarified out experimental strategy
- checked and amended where needed the equations highlighted as in error

We have however not substantially modified our methods, but we believe with the revision we have justified our purpose and value of the analysis performed and the tool presented for other researchers to explore their own data.

We outline our responses and revisions that are included in a revised manuscript below.

**Major comments**

(1) I see a few possible careless mathematical errors and inappropriate assumptions, which could have been avoided. For example,

1. The expansion given in equation 20 is wrong.  $(1+x)^{-2}=1-2x+3x^2-4x^3+5x^4+\dots$

2. Please check Eq 9. For example, according to https://en.wikipedia.org/wiki/Propagation_of_uncertainty,  if $\sigma_f = a/b$,

$$\sigma_f \approx \sqrt{\sigma_a^2 \left(\frac{1}{b}\right)^2 + \sigma_b^2 \left(\frac{-a}{b^2}\right)^2}.$$

then

3. In section 3.5, equation 19, you assume that the temperature values correspond to equispaced sensors and thus errors in the numerator vanishes. However, the spacings are not equal anymore due to the

random shifts (both in a real experiment and your simulations). The errors in the numerator must be considered.

We apologize for the equation errors, which are in the manuscript rather than the calculations and so do not affect the general results of the paper.

Section 3.5 has been rewritten and partially removed, which involved also removing equations 19 and 20. The published python tool uses the equations for unequal spacing of thermistors, so the equation shown is not relevant and has been revised. Furthermore, in revising the manuscript we noticed that Eq. 9 is not used further in the manuscript, and it was also removed. We have checked all other equations in the manuscript and cross-checked those behind the interactive tool and figures produced to ensure no remaining errors.

(2) A couple of recent publications (Laha et al. & Petersen et al.) went beyond the assumptions of a homogenous, source-free, purely conductive heat flux, as being considered here. You need to provide more compelling an argument about the motivation behind and significance of the present study than you do in L107. (Of course it is a different matter, if you were to actually analyse the existing data of thermal diffusivities reported in the literature, but that is not something you attempt here.)

We agree that the forward model and inclusion of a 2 layer consideration presented in Laha and others (2022) is demonstrated to outperform the Conway and Rasmussen (2000) method for cases where one is trying to calculate apparent thermal diffusivity from data in vertically varying debris and with irregularly spaced thermistors, however we maintain their is value in our analysis as:

- Much of the historical data upon which literature sourced values of debris thermal conductivity use the Conway and Rasmussen (2000) method, and understanding the limitations on the inter comparability of this data is important.
- Implemented correctly in the field the Conway and Rasmussen (2000) method still holds the potential to determine relevant thermal conductivity values representative for the bulk debris properties, as required for surface energy balance modelling. This is evidenced in Laha and others (2022) by the identical performance of this model to the newer methods for equal spaced thermistors and homogenous debris - with the implication that the method remains applicable to debris temperature profiles that confirm to these conditions, which is indeed how early studies applied it.
- We are interested in capturing the thermal conductivity throughout the debris layer to generate input parameterization for melt models and/or to identify stratigraphy of properties and/or non-conductive processes in the debris layer. While the method of Laha does account for two layers, it does not yield thermal conductivity values for each layer. This would be needed for application to a multilayer surface energy balance model where fluxes are driven from the surface rather than an internal temperature measurement.

To address this in the text, we introduce a section stating our purpose:

*"As the existing, and limited, sets of field data used to provide generalized values for the effective thermal conductivity of unmeasured glacier sites have been analysed based upon the simple Conway and Rasmussen method, rather than the later methods developed by Laha and others, there is value in a deeper exploration of the limitations to interpreting values derived by this method, in particular to better understand the meaning of comparison between sites at regional and global scales (e.g. Rowan et al., 2021; Miles et al., 2022). For example, the measurement parameters for temporal or spatial sampling intervals, thermistor spacings, and debris depths used in the application of the standard method presented by Conway and Rasmussen (2000) are selected ad hoc and differ from measurement site to measurement site (e.g. Juen et al., 2013; Chand and Kayastha, 2018; Rowan et al., 2021), and uncertainty estimates associated with this are missing. This means that baseline literature values that are subject to onward use on the literature may be differently influenced by sensor, installation and numerical truncation errors. This study explores the effect of the chosen temporal and vertical spatial temperature sampling interval and other systematic measurement errors originating in the measurement setup on the derived thermal diffusivity values. To explore the capabilities and limitations of this approach we apply this method to artificially generated data with a known value of thermal diffusivity, which allows us to individually quantify systematic and statistical errors by error source. We additionally present an online tool to allow interactive analysis of these combined errors for a given dataset and a best practice guideline on how to minimize the systematic errors inherent in the methods of Conway and Rasmussen (2000)."*

(3) While the space- and time-discretisation steps are important for setting up actual measurements, it is difficult to judge whether your analysis can actually lead to useful insights about real experiments, due to the idealisations involved in your study design. You have totally ignored the sources that are present due to the horizontal inhomogeneities and temperature gradients, water content, advected heat, latent heat transport, convection etc, and vertical variation in Kappa. If one were to incorporate the noise due to these effects in the forward model, which are present in the real system anyway, will your results hold?

We agree that in order to apply this method correctly in the field a site must be chosen that is minimally affected by these processes, and even then the full temperature profile should be investigated to understand the potential variation of debris properties and non-conductive processes being applied.

Nevertheless, where one can identify 'well behaved' sections of the debris temperature profile that conform to the underlying linearity of the heat equation for conductive systems, the approach remains valid. This is inline with the traditional application of the method which emphasised finding optimal conditions for thermal diffusivity determination in order to extract reasonable values of thermal properties for subsequent use in generalised energy balance modelling.

Once an optimal site for analysis is found the systematic error sources revealed by our study still apply and these should be additional considered in the ideal installation of an array of thermistors.

(4) Since you are solving the forward problem numerically, it is easy to create an ensemble of experiments where all the parameters and variables in your model are perturbed by appropriate space and time dependent noise, and the corresponding mean values are drawn from a distribution. If one does that, then all the lines in your plot will have an associated uncertainty band. When such uncertainty is considered, the differences between different curves that you have discussed throughout your manuscript may become insignificant. Till you do this exercise and demonstrate that your results/conclusions are robust against such the inherent variability and measurement noise, your conclusions are not on a firm footing.

Our goal is to show the fundamental behaviour and consequences of the choices of how to instrument and analyse thermistor data that are embedded within all the real world complexity - which we agree can be large. In fact the analysis you describe was already done in Laha and others (2022) for hypothetical data and, as the equations for the CRh model used in that study are the same as the ones we use, we see less community value in repeating that, and prefer to to focus on theoretical cases to understand the underlying behaviour. Nevertheless to address this comment we now add text to clarify the magnitude of the error sources we tackle in our analysis in comparison to the variability seen in published field observations to set our findings in context.

**Other comments**

I think there is scope and need for improving the writing. There are several sentences which either lacks justification, or are vague, or even incorrect. Also, it may be better to avoid subjective discussion when you set up/introduce the equations and symbols, eg, at the beginning of your methods section. Please just state the standard definitions, and provide units. Please revise/reconsider the sentences/phrases listed below.

L18: The regional-scale debris-covered effect was discussed in several papers eg, Scherler et al, Nature Geosci., 2011, Banerjee & Shankar, 2013. Even though Hock et al. 2019 ignored such effect, it was not previously thought to be unimportant - only there was ready-made way to incorporate it in large-scale models.

L33: Not sure how attenuation of daily signal controls heat flow. Of course, it is a consequence of the diffusive evolution of $T(x,t)$.

L34: what is "thermal instability" in a conductive system?

L47: What processes? "The supply of melt energy" can never be "represented" by an effective thermal conductivity.

L57: Despite the long, general introduction, the question addressed in this paper is not motivated at all.

L87: Does this paragraph belong to methods? Seems to be more suitable for the introduction section.

L135: please demonstrate that it is enough to consider the intercept, to incorporate all the errors/uncertainties mentioned in major comment 3 and 4, for example.

L197: Limiting values are still well defined.

L206: Please provide mathematical justification or some relevant reference where such justification has been provided, for this average method (method 2). What you are calling skipping, is just a standard finite-difference method as explained in any textbook.

We have substantially reorganised, restructured and re-written the text so in that process all the problematic points below have been removed or corrected. In particular, we have clearly stated our goals, reduced and reorganised the introduction, ensured that methods are contained in the appropriate section, and added a conclusion that covers aspects of the general application of this method in both numerical and real-world terms. We are confident that this takes care of the concerns about the quality and clarity of the writing as well as the highlighted points above, but specifically we note re. L135: we have no systematic error in our artificial data but we also add a comment that for real world data the intercept contains error terms that are not represented in our analysis and re. L206: We would advise temperature sampling in time, but this averaging method was used for all field sampling of thermistor datasets from the Khumbu glacier in Rowan and others (2020) - as such we include it in our analysis to allow people to understand the implications of that. We add an explanation of this reason in the text.

---

## Author Comment (AC2)

**Review 2**

I like the idea of this paper. I believe not enough attention is placed on error slinked to both study design and data collection, and numerical approaches. This paper discusses both these issues extensively concerning the calculation of the thermal diffusivity, a key parameter in calculating the diffusion heat flux to the surface, and the subsequent ice melt. Specifically, the influence of temporal and spatial temperature sampling intervals in the data collection, as well as the influence of truncation errors in the calculations, is explored along a gradient of idealized data to observed data. While this approach has limitations, like assuming a homogeneous debris layer, I think drawing attention to often ignored sources of uncertainty is valuable.

However, there are some flaws in the manuscript that I think should be addressed. First, the manuscript would benefit from closer attention to the writing. In many places, the writing is highly informal and imprecise. While I value directness and simplicity to convey a clear message, in some cases, the manuscript reads more like a blog post, with vague statements and a lack of structure.

We thank the reviewer for their review and critical comments about the manuscript and work. In particular the comments about the presentation which motivated a substantial overhaul and re-writing of the text, through which we have addressed all the comments raised.

The text would benefit from being streamlined to allow for a clearer flow.
There are also some sections of methods that read like introduction and discussions that also read like methods.
The discussion lacks a section on the limitations of this study
There is no conclusion, simply a long list of broad best practice guidelines that are very general.

We have restructured the manuscript and by doing so have improved the flow and also the division of material across the sections, we have stated the aims in a new section and discuss the limitations of the study and the scope of this study, as well as adding a conclusion that covers aspects of the general application of this method in both numerical and real-world terms.

Citations are formatted wrong and many places and general lack of commas throughout the text.

Overall, I found the text hard to follow and found it hard to see the key points in the study. Therefore, despite my support of this study, I think it needs a lot of clarification and rewriting before it can be published.

We apologise that the readability was not up to standard first time round, and have now re-written the whole manuscript and improved the readability and logical flow of the manuscript. We have checked citation formatting and grammar.

---

## Author Comment (AC3)

**Review 3**

The authors present a theoretical experiment to estimate thermal diffusivity within the debris layer of a debris-covered glacier. The topic of thermal properties within debris-cover is timely and relevant given the many other studies that have been published in the past few years. Overall, the paper is well thought through and lays out the structure and approach used. I think that the writing style could be improved to be more formal as is typical in a journal article, and attention to detail is needed for many sentences throughout. There are many minor edits that I suggest throughout, and in general the text should be made much clearer as it is difficult to follow in places. The entire paper should be read with an eye towards grammar and citation formatting, as there are many (beyond the comments I made) that are incorrect. I think the overall suggestions are useful for future work, and should be an informative paper, but the structure and cleanliness of the paper needs to be improved before publication.

We thank the reviewer for their review and critical comments about the manuscript and work. We have substantially restructured and rewritten the manuscript, improving the organisation and flow, and ensuring no repetition. We have added a clear aims and experiments section and checked all language and formatting carefully.

I think that the paper clearly states the justification to focus its analysis on the Conway and Rasmussen (2000) paper, which I feel accounts for many of the comments of a prior reviewer. To address this, it might be useful to provide more of a comparison between the newer Laha et al (2022) paper and the Conway and Rasmussen (2000) paper to further support the need for this study. I think that the value of this paper comes from its theoretical approach instead of using entire seasons of existing data from other papers.

We agree in the that value of our study, but in response to reviewer 1 have added a clearer statement of purpose to address this point:

"*As the existing, and limited, sets of field data used to provide generalized values for the effective thermal conductivity of unmeasured glacier sites have been analysed based upon the simple Conway and Rasmussen method, rather than the later methods developed by Laha and others, there is value in a deeper exploration of the limitations to interpreting values derived by this method, in particular to better understand the meaning of comparison between sites at regional and global scales (e.g. Rowan et al., 2021; Miles et al., 2022). For example, the measurement parameters for temporal or spatial sampling intervals, thermistor spacings, and debris depths used in the application of the standard method presented by Conway and Rasmussen (2000) are selected ad hoc and differ from measurement site to measurement site (e.g. Juen et al., 2013; Chand and Kayastha, 2018; Rowan et al., 2021), and uncertainty estimates associated with this are missing. This means that baseline literature values that are subject to*

*onward use on the literature may be differently influenced by sensor, installation and numerical truncation errors. This study explores the effect of the chosen temporal and vertical spatial temperature sampling interval and other systematic measurement errors originating in the measurement setup on the derived thermal diffusivity values. To explore the capabilities and limitations of this approach we apply this method to artificially generated data with a known value of thermal diffusivity, which allows us to individually quantify systematic and statistical errors by error source. We additionally present an online tool to allow interactive analysis of these combined errors for a given dataset and a best practice guideline on how to minimize the systematic errors inherent in the methods of Conway and Rasmussen (2000)."*

We also now discuss the relative merits of the approaches in the discussion section, including the applications or use cases for which each is best.

The methods used are well outlined overall and supported by the figures and online tool provided in the paper. There are a couple paragraphs in the methods that could be more useful in the introduction or discussion (L87, L110) Please see my in-line comments regarding minor edits to certain paragraphs and sentences.

We have reorganised the sectioning, and moved these parts, and streamlined the introduction in places.

It is important for readers to understand that this is a theoretical model and will of course not account for every eventuality that might occur in the "real debris" layer on a glacier, but I think a major benefit of this study is the online tool to explore the errors interactively. This tool was easy to use and provides the theoretical knowledge to explore thermal diffusivity within a debris-layer. The assumptions that are made regarding density, conduction, and specific heat capacity are reasonable considering the constraints of a numerical study. The results show that this model holds up against specific instances of real data which helps validate the model being presented here. However, there should be more in the discussion or conclusion about the limitations of this study and more specific reasoning for the best practice guidelines that it provides.

Thank you, we emphasise this in the new section explaining the experiments performed and also discuss the limitations more fully, making clear distinctions between understanding the numerical behaviour in response to theoretical sampling and implementations. We also emphasise the tool more within the discussion text.

As it is intended, this paper focuses its analysis on the Conway and Rasmussen paper and identifies limitations and error sources based on that method of estimating thermal diffusivity values. This paper provides value to the literature because of how frequently the Conway and Rasmussen method is used and oftentimes, without deeper consideration for its error sources and limitations. This

paper does not aim to analyze other new methods as discussed in Laha et al., and Petersen et al., and I think this is okay due to the relative lack of use of these newer methods. While the guidelines and suggestions that this paper makes are based on a theoretical model, there needs to be more consistency in the field-based methods used in this discipline, so this paper provides some suggestions for how to do that.

Thanks, we agree, and have aimed to clarify this with a better purpose statement within the manuscript.

The best practice guidelines provided are helpful considerations for future field studies that aim to measure thermal diffusivity within the debris-layer. The debris-covered glacier literature needs to have more consistent methods of measuring thermal diffusivity to compare findings across different field sites. The guidelines this paper presents should be used in the future by other field-based studies and these future studies should consider the limitations and error sources that are discussed here. Please just add to these guidelines and explain the reasoning for these guidelines in a clear and structured manner.

We have expanded upon our reasoning, and in particular highlight the reasons for contrasting recommendations for different purposes (e.g. Laha and others (2022) seek a method to best determine sub debris ablation rate from thermistor measurements directly, while we explore methods used to determine thermal conductivity values for application in generalised surface energy balance models of ablation beneath debris cover).

**Line by Line comments – also in line on PDF.**

L57: Specify vertical spacing and provide slightly more reasoning for why you are exploring these variables in the sentences before. Also discuss why horizontal spatial variability is not focused on in this study.

We clarify this and highlight the potential for real-world fluxes to differ from the idealised 1D heat equation, restating that we aim to show the consequences only of the calculation implementation, not the degree to which the real world meets the assumptions of the method.

L87: This paragraph could be better in the intro or discussion sections.

Moved, and streamlined

L93: Grammatical issues. A period or comma is needed, and the spacing is strange.

Addressed in the full rewrite of the manuscript.

L103: Be more specific in terms of "they."

Addressed in the full rewrite of the manuscript.

L103-109: Also adding another sentence here to improve the justification of this approach would help convince readers this is a valuable approach you are taking. Provide more comparison from Laha et al. and Petersen et al.

Agreed, we provide more information on the benefits of each method and their appropriate usage, in principle and in existing literature.

EQ 3, 4, 5: The O in these equations is not defined and is not consistent across these three Eqs.

These equations were removed when this section was moved into a more streamlined introduction section.

L177: Figures need to be cited in line, and when cited need to be clearly referencing that given figure.

Done, figure captions and in-text citations of them have been checked throughout.

Figure 3 caption: data 3, 4, 5 are not shown on the graphic – I think I know which data you are referring to, but please clarify and provide the same description as the values in the figure.

Done, figure captions and in-text citations of them have been checked throughout.

L182: Another figure citation missing.

Done, figure captions and in-text citations of them have been checked throughout.

Figure 4 caption: "timeseries" will need a space.

Done, figure captions and in-text citations of them have been checked throughout.

L184: Figure 6 is being referenced here and it seems a bit out of order if you are indeed citing that figure. Please make sure every figure is referenced.

Done, figure captions and in-text citations of them have been checked throughout.

L206-208: Provide a little more clarity on these two resampling methods. Method 1 is clear, but method 2 is less so, provide more detail here to avoid confusion.

Thanks, we clarify that it is averaging over a time interval and explain that we choose to investigate that as in some of the reported field data temperatures (e.g.

Rowan et al, 2021) for Khumbu glacier were collected as time averages rather than samples.

L208: Figure reference missing.

Done, figure captions and in-text citations of them have been checked throughout.

L218: Make this entire paragraph more clear.

Paragraph has been rephrased

L272: "purly" to purely

Done
Eqs. 16-20: Why are these equations in the results? Shouldn't these be in the methods section and discussed there?

Removed or moved.

L290: This paragraph in discussion needs to be re-written or at least made more clear. It is confusing and doesn't read smoothly.

Paragraph has been rephrased

L316: remove comma after "true, "
Done

---

## Author Response (AR2)

**Response to second reviews**

We thank both reviewers for their time in considering our paper again, and are happy to hear that the paper has improved. As the reviews essentially have the same concerns we treat them together here, leading with the comments of Reviewer 1.

**Reviewer 1** asked for us to add random temporally constant but space variant noise in the forward model to set our relative error analysis in the context of likely uncertainty associated with real world debris cover, specifically referring to the ways that natural debris might not meet the assumptions of the method we are testing. In the end, we still prefer to not do this beyond what we have already done in the paper, for the reasons explained below. We have however included further analysis exploring the role of noise within the method to highlight the limits at which the method breaks down and is no longer applicable, which has resulted in 2 new figures (Figures 10 and 11).

Firstly, the thermistor location accuracy experiment that we perform already includes the analysis requested - a temporally constant but space variant locational offset sampled randomly from a Gaussian distribution. This case represents both 'poorly known location' and also non-conformity to the assumptions of the method, because in effect this resampling applies the method to a temperature profile that is no longer that of a purely conductive system. What we see is that, as the Gaussian error is symmetrical, the mean of the ensemble returns the target Kappa value, aside for impractically close thermistor spacing (see Figure 9 in our paper). The issue with adding Gaussian noise in general to the forward model to represent all the potential error of non-conductive processes, is that the nature of a distribution that represents the published real world debris cases is impossible to determine meaningfully, as it could be very small for cases that closely approximate the assumptions of the method, or very (indefinitely) large if the sampling site is poorly chosen and suffers from inhomogeneity and significant non-conductive processes. So we would be adding a potentially unlimited uncertainty to our analysis. Instead, by isolating only methodological errors and error behaviour we can make specific recommendations for deployment and analysis strategies. If we add noise to the target Kappa then we can no longer determine the source of error produced. Our study follows the structure of synthetic experimentation performed in Laha et al, 2022 to assess the numerical error of uneven thermistor spacing, expanding that analysis to a wider range of numerical errors related to measurement choices. Laha et al., 2022 also did not apply Gaussian noise to their synthetic cases, rather only to the field datasets, which we agree seems appropriate and desirable. By isolating error sources independently our approach allows us to identify that the error response of adding Gaussian noise to the measurement position varies with vertical sampling interval, from that we can see that while the Gaussian noise added to real temperature data from sites 4 and 7 on Santopath glacier in Laha et al (2022) show similar error response, this would be expected as both of the sites analysed use a sampling interval of 6cm, and this error response might however, not be representative for sites on

the same glacier with different vertical sampling interval.

Secondly, as the point of our study is to identify meaningful best practice guidelines for measurement set up, we must focus on the methodological errors inherent in the Conway and Rasmussen method. For this it is only fair to test the method in cases for which it is valid - i.e. in which the debris is homogeneous and closely approximates a purely conductive system. Laha et al 2022 already nicely demonstrate the weaknesses of applying this method when these assumptions are not met, and provide an alternative method that outperforms the one we are testing in those circumstances, and other studies highlight how to identify when these conditions are not met in the real world datasets (Nicholson and Benn, 2013, Petersen et al, 2022). We have included an assessment of statistical noise to showcase the behaviour of the method when applied to spatial-temporal gradients that can no longer be meaningfully assessed by the method to yield Kappa values. As the vast majority of published data points are based on the original Conway and Rasmussen method, we still feel it is useful to present its associated numerical errors alongside a tool aimed to allow colleagues to easily determine an optimal strategy for future field deployments.

**Reviewer 2** suggested that we should follow the uncertainty assessment of Laha et al, 2022. However we note that these authors only applied the type of Gaussian noise uncertainty to their observed field datasets, which we agree is entirely appropriate and desirable. As we have done in our study, for their analysis of *synthetic cases* to assess the error associated with uneven thermistor spacing and choice of method , Laha et al., 2022, also did not apply noise, presumably because they also wanted to preserve the known target Kappa value in order to be able to assess methodological performance at recovering it.

To address the reviewer points within the manuscript we now:

- Emphasise that care should be taken to apply the analysis only to subsets of the data which can be shown to be predominantly conductive. These conditions have been established through previous work to be best met in a matrix-supported diamict, during stable meteorological conditions within the established ablation season, and that data from these conditions will yield the most reliable result of apparent thermal diffusivity for use in models.
- Emphasise throughout the paper why we wish to test only the inherent methodological errors of this method that has produced the vast majority of available published data points for debris thermal properties.
- Include an analysis of Gaussian statistical noise on the temperature timeseries to demonstrate the limits of the method in cases where the spatio-temporal gradients are degraded to the point where they are no longer interpretable by the method being tested. These results are shown in new Figures 10 and 11.

- Removed the background shading for the figures that had been used to distinguish between skipping and averaging temporal resampling, as requested by Reviewer 2.

Additionally we also:

- Stress that the examples we show are illustrative of the relevant parameter space that can be explored in its entirety with the interactive tool provided - as a means to more strongly promote the use of the provided tool to determine optimal field sampling strategies prior to sensor deployment.
- Relatedly we have added more information on the details of each case shown, as in places this was incomplete in the former manuscript.
- Include reference to the recently published DebDab dataset (Fontrodona-Bach et al., 2025), which includes the most complete database of published and non-published debris thermal conductivity values.

These changes entailed the inclusion of some additional text compared to the previous version, which also included some editorial

---

## Author Response (AR3)

We thank the editor, Ben Marzeion, for his support during the revision process and for accepting our revised manuscript. We have addressed all the minor technical/editorial issues that were raised and look forward to the paper being published.